# Contrasting effects of SARS-CoV-2 vaccination vs. infection on antibody and TCR repertoires

Jasper Braun[1], Elliot D. Hill[1], Elisa Contreras[1], Michie Yasuda[1], Alexandra Morgan[1], Sarah Ditelberg[1], Ethan Winter[1], Cody Callahan[1], Gabrielle Mazzoni[1], Andrea Kirmaier[1], Ghee Rye Lee[2¤], Hamid Mirebrahim[3], Hosseinali Asgharian[3], Dilduz Telman[3], Ai-Ris Y. Collier[4,5,6], Dan H. Barouch[4,6,7,8], Stefan Riedel[1,4], Sanjucta Dutta[1], Florian Rubelt[3], Ramy Arnaout 🔾[1,4,9]*

1 Division of Clinical Pathology, Department of Pathology, Beth Israel Deaconess Medical Center, Boston, Massachusetts, United States of America, 2 Department of Radiology, Beth Israel Deaconess Medical Center, Boston, Massachusetts, United States of America, 3 Roche Sequencing Solutions, Pleasanton, California, United States of America, 4 Harvard Medical School, Boston, Massachusetts, United States of America, 5 Department of Obstetrics and Gynecology, Beth Israel Deaconess Medical Center, Boston, Massachusetts, United States of America, 6 Center for Virology and Vaccine Research, Beth Israel Deaconess Medical Center, Boston, Massachusetts, United States of America, 7 Ragon Institute of MGH, MIT, and Harvard, Cambridge, Massachusetts, United States of America, 8 Department of Medicine, Beth Israel Deaconess Medical Center, Boston, Massachusetts, United States of America, 9 Division of Clinical Informatics, Department of Medicine, Beth Israel Deaconess Medical Center, Boston, Massachusetts, United States of America

¤ Current address: Ohio State University College of Medicine, Columbus, Ohio, United States of America
* rarnaout@bidmc.harvard.edu

## Abstract

Antibodies and helper T cells play important roles in SARS-CoV-2 infection and vaccination. We sequenced B- and T-cell receptor repertoires (BCR/TCR) from the blood of 251 infectees, vaccinees, and controls to investigate whether features of these repertoires could predict subjects' SARS-CoV-2 neutralizing antibody titer (NAbs), as measured by enzyme-linked immunosorbent assay (ELISA). We sequenced recombined immunoglobulin heavy-chain (IGH), TCRβ (TRB), and TCRδ (TRD) genes in parallel from all subjects, including select B- and T-cell subsets in most cases, with a focus on their hypervariable CDR3 regions, and correlated this AIRRseq data with demographics and clinical findings from subjects' electronic health records. We found that age affected NAb levels in vaccinees but not infectees. Intriguingly, we found that vaccination and infection are associated with longer non-productively recombined IGHs, suggesting an effect that precedes clonal selection. We found that TRB repertoires' binding capacity to known SARS-CoV-2-specific CD4+ TRBs performs as well as the best hand-tuned approximate or "fuzzy" matching at predicting a protective level of NAbs, while also being more robust to repertoire sample size and not requiring hand-tuning. The overall conclusion from this large, unbiased, clinically well annotated dataset is that B- and T-cell adaptive responses to SARS-CoV-2 infection and vaccination are surprising, subtle, and diffuse. We discuss methodological and statistical challenges faced in attempting to define and quantify such

**Data availability statement:** All raw data for this study can be found at: https://data.entrop. ai/cdr3/data/index.html.

**Funding:** This work was supported by the National Institutes of Health (R01AI148747 and R01AI148747-SI to RA), the Massachusetts Life Sciences Center (RA), the Extreme Science and Engineering Discovery Environment (RA), and the San Diego Supercomputer Center (RA).

**Competing interests:** HM, HA, DT, and FR are current or former employees of Roche Molecular Systems Inc. JB, EDH, EC, MY, AM, SD, EW, CC, GM, AK, A-RYC, DHB, SR, SD, and RA have no conflicts of interest to declare.

strong-but-diffuse repertoire signatures and present tools and strategies for addressing these challenges.

## Introduction

Since COVID-19's emergence, there has been great interest in identifying antibody (BCR) and T-cell receptor (TCR) gene sequences specific to SARS-CoV-2. The pandemic presented a high-profile opportunity to test the extent to which TCRs and BCR sequences against respiratory viruses would be public, i.e., appearing across many different individuals, or private, present in only one or a few individuals. New SARS-CoV-2-specific BCRs could form the basis for new treatments. Understanding how to identify and characterize commonalities in such a setting could help evaluate the viability of new diagnostics based on adaptive immune-receptor repertoire sequencing (AIRRseq) [1]. A number of studies have succeeded in identifying public immunoglobulin heavy-chain (IGH) and TCRβ (TRB) sequences. However, in part due to exigencies and constraints imposed by the pandemic, and in part because it was impossible to know a priori what study size would be adequate to identify public sequences comprehensively in COVID-19, many of these studies involved relatively few individuals. Small sample sizes have known limitations for AIRRseq studies. Because small samples may not be representative of larger populations, results on small samples may not generalize. Small studies may be insufficiently powered to detect subtle patterns. And the smaller the sample size, the more likely it is that random fluctuations in the data—literally, the luck of the draw—will produce results that appear to be statistically significant but do not reflect underlying relationships. Moreover, most studies investigated only BCRs or only TCRs, but not both in the same cohort, despite the importance of both antibody and T-cell responses in SARS-CoV-2 infection [2,3]. To our knowledge only two previous COVID-19 studies [4,5] have sequenced TCR from γδ T cells, a little- studied subset that may be important in mucosal antimicrobial immunity [6,7]. How SARS-CoV-2 virus or vaccine exposure affects different B- and T-cell subsets (IgM-positive vs. IgM-negative B-cells, CD4-positive vs. CD8-positive T cells) has also been insufficiently explored [2,3].

To our knowledge previous studies have identified 20 IGH V genes to be enriched in sequences produced during various immune responses to SARS-CoV-2 [8–16]. Given that human genomes encode 54 IGH V genes [17], collectively these studies implicate 37% of all V genes in the response to this single viral exposure, indicating that the SARS-CoV-2 response is either quite broad within individuals, quite heterogeneous among individuals, or both. There is no obvious reason to think each V gene would contribute to only a single SARS-CoV-2-specific recombined IGH sequence, much less a single IGH:immunoglobulin light chain (IGL) pair; therefore, collectively these studies also suggest that the IGH response to SARS-CoV-2 might account for a significant fraction of a given repertoire, a possibility that requires more comprehensive AIRRseq investigation. Note that studies that compare only a single non-control cohort to a control cohort cannot distinguish between features (clones, motifs, genes, CDR3 lengths, etc.) that signify a disease-specific vs. a general immune response.

Regarding TCRs, one study [18] searched repertoires of 140 COVID-19 patients and another 140 pre-pandemic (and therefore unexposed) controls for the presence of each of 1,267 TRB sequences that had independently been shown to recognize epitopes of the SARS-CoV-2 spike protein. These authors showed that while the presence of some of the TRB sequences in almost all of the repertoires suggested a public response to SARS-CoV-2 infection, the fraction of the repertoire that matched the query sequences was similar between infectees and controls. These authors also looked for SARS-CoV-2 specific TRBs in the brain tissue of COVID-19 patients, since T-cell infiltration of the brain, an organ otherwise seldom infiltrated by T cells, is known to occur during COVID-19 infection [19–21]. The 68 TRBs they identified were found in 40% of COVID-19 repertoires vs. 17% of pre-pandemic controls. This suggests significant enrichment even as the majority of individuals with COVID-19 lacked these sequences and a significant minority of controls (1 in 6) had them despite these control samples having been collected before the pandemic (which has been observed in other contexts [22], perhaps indicating cross-reactivity with previously circulating coronaviruses, which are very common in human populations [3]).

In addition, one study [23] identified a large database of SARS-CoV-2-specific TRB sequences as being shared among infectees and unenriched among healthy controls. Subsequently, another study [24] sequenced repertoires of individuals 0 and 4 weeks after vaccination with the Oxford-AstraZeneca COVID-19 vaccine AZD1222 or the meningococcus vaccine MenACWY and matched the database sequences to these repertoires. An increase of database sequences among AZD1222-vaccinee repertoires but not MenACWY-vaccinee controls was seen between the two timepoints.

Heterogeneity in clinical settings across studies complicates the interpretation of private vs. public responses, for several reasons. First, there are important antigenic differences between exposure and vaccination. This is especially true for the mRNA vaccines, which immunize subjects with only spike protein, in contrast to the full complement of SARS-CoV-2 proteins to which infectees are exposed. Second, demographics may play a role.

For example, it has long been recognized that individuals respond differently to vaccines by age, with older individuals generally mounting less-robust and shorter-lasting responses as measured by ELISA [25–27]. Other clinical features are also known to affect the adaptive immune response to infection and vaccination, including immunosuppressive conditions such as organ transplant or cancer therapy as well as metabolic disease [28]. Third, studies from different periods of the pandemic likely measured responses to different strains. Fourth, the signal or signature detected may differ depending on whether the controls were healthy, which might result in detecting generalized responsiveness (e.g., bystander activation), or were instead presenting with a non-COVID illness, making any signal/signature more likely to be specific for COVID-19. Fifth, exposure, whether to replicating virus or to an inactivated or subcomponent vaccine, may not be as clinically relevant as whether a substantial NAb response was mounted. This is because NAbs are a marker of protection in SARS-CoV-2 [29–31]. (T-cell-mediated immunity may also play an important role [32]). And sixth, accessing clinical annotations from electronic medical records can be challenging [33]. As a result, the effect of clinical heterogeneity in AIRRseq studies in the setting of COVID-19 has been under-explored to date.

In all, the work above supports the view that there are commonalities in IGH and TRB at the gene and sequence level in response to SARS-CoV-2 infection; however, the nature of the signature is not well understood. One open question is to what extent infection affects antibody and TCR repertoires as a whole vs. enriching specific clones within it. One can refer to these ends of the continuum of possible effects as "diffuse" vs. "precise." From previous work on repertoires, "diffuse" features include CDR3 length, the frequency of usage of specific V or J genes, and repertoire diversity as measured any of several ways (richness, Shannon entropy, Simpson's index, or their Hill-number equivalents) [34]. At the other extreme, the most "precise" features are the frequency of clones with specific sequences. Between these extremes is a set of features that includes approximate or "fuzzy" matching of sequences [35] and other clustering methods [36,37]. This middle ground has been less explored. Recently the concept of binding capacity has been developed to measure the fraction of a repertoire that is "like" a given query sequence in terms of target specificity (weighting the repertoire by the predicted dissociation constants of its constituent antibodies or TCRs and by their sequence frequencies). Whether or how binding capacity might be affected by SARS-CoV-2 infection and/or vaccination is unknown.

Given this background, we sought to investigate the effects of SARS-CoV-2 infection and vaccination on both antibody and TCR repertoires in a large clinical cohort, with attention to major B- and T-cell subsets where possible, using NABs via ELISA as a functional readout, with a special focus on diffuse repertoire features and how they compare to both more traditional features and to clinical correlates.

## Results

### NAbs vary with exposure, age, and immune status

Using immunoPETE (Roche; research use only) [5], we deep-sequenced IGH, TRB, and TRD from the blood of 251 individuals: 36 vaccinees, all of whom had received an mRNA vaccine, 145 infectees, 53 healthy controls, and 20 with unknown SARS-CoV-2 exposure status. Three individuals belonged to both the vaccinee and infectee groups. Forty-seven subjects were considered immunosuppressed and the remaining 204 immunocompetent. Blood samples for 129/145 (89%) infectees were within 6 months of the most recent positive PCR test on record and 121/145 (83%) were ≥7 days from the presumed most recent infection date (assuming a mean of 4 days from exposure to testing). Fig S1 in S1 File presents a summary of the timeline and sequencing yield. Tables S1 and S2 in S1 File present demographics and relevant comorbidities for the different cohorts. We measured plasma NAbs against SARS-CoV-2 spike for 237/251 subjects. Fig 1 presents a summary of the measured NAbs concentrations by cohort, immunosuppression status, and age.

NAbs were undetectable in some infectees and one vaccinee (Fig 1). The odds of producing NAbs were significantly higher in immunocompetent subjects compared to immunosuppressed subjects (OR=3.9, $p_c$ = 0.008—note, all p-values in this work have been corrected for multiple- hypothesis testing; we write $p_c$ to indicate this). Odds were also significantly higher in the infectees (OR=5.8, $p_c$ < 0.001) and vaccinees (OR=4.7, $p_c$ = 0.034) compared to controls. Age was not significantly associated with NAbs titer ($p_c$ = 0.801) and therefore age was excluded from most of the subsequent models (below).

Among the subjects who did produce detectable NAbs, NAb concentration was notably higher in the vaccinated and infected groups compared to the control group (Fig 1a). The relationship between concentration and age was more complex and depended on the cohort (significant age × cohort interaction). Age affected titer only in vaccinees, with NAbs being lower in older individuals: above age 65, individuals had higher NAbs with infection than vaccination (Fig 1a). Meanwhile, age did not affect NAbs in the control or infected groups. We found no significant effect of immunocompetence on NAbs in subjects who had non-zero NAbs.

### Vaccination is associated with shorter IGH CDR3s in productive joins

The characteristic (e.g., mean or median) length of CDR3s is known to vary during development and in response to various exposures, at least in productively recombined IGH genes, a.k.a. "productive joins" [38]. Because only productively recombined IGH genes can be expressed as (BCR) proteins, such differences are generally considered evidence that the B cells that express them are selected for having, e.g., longer CDR3s. We found that vaccinees had shorter IGH CDR3s than controls ($p_c$ = 0.024; Fig 2a) or infectees ($p_c$ = 0.0046; Fig 2b), indicating a repertoire-wide difference in the B-cell response to vaccination vs. infection (Table S3 and Figs S2-S4 in S1 File).

The length of IGH CDR3s depends on the lengths of the constituent IGHV, IGHD, and IGHJ genes, as well as the number of non-templated (N) and palindromic (P) nucleotides inserted at the junctions between them [39]. Annotating IGHD and distinguishing mutated/truncated IGHD sequence from N and P sequences is challenging due to insertions, chewbacks, and somatic hypermutation. However, IGHV and IGHJ can be annotated reliably, and so we tested whether the overall differences in CDR3 length were attributable to differences in the use of longer vs. shorter IGHV and IGHJ genes.

We grouped IGHV genes by the number of amino acids that their germline contributes to CDR3s, and similarly for IGHJ genes. The 54 IGHV genes hard-coded as part of the human germline contribute either 3 or 4 amino acids to the CDR3, depending on the gene. We found that vaccinees generally used more of the IGHV genes that contribute 3

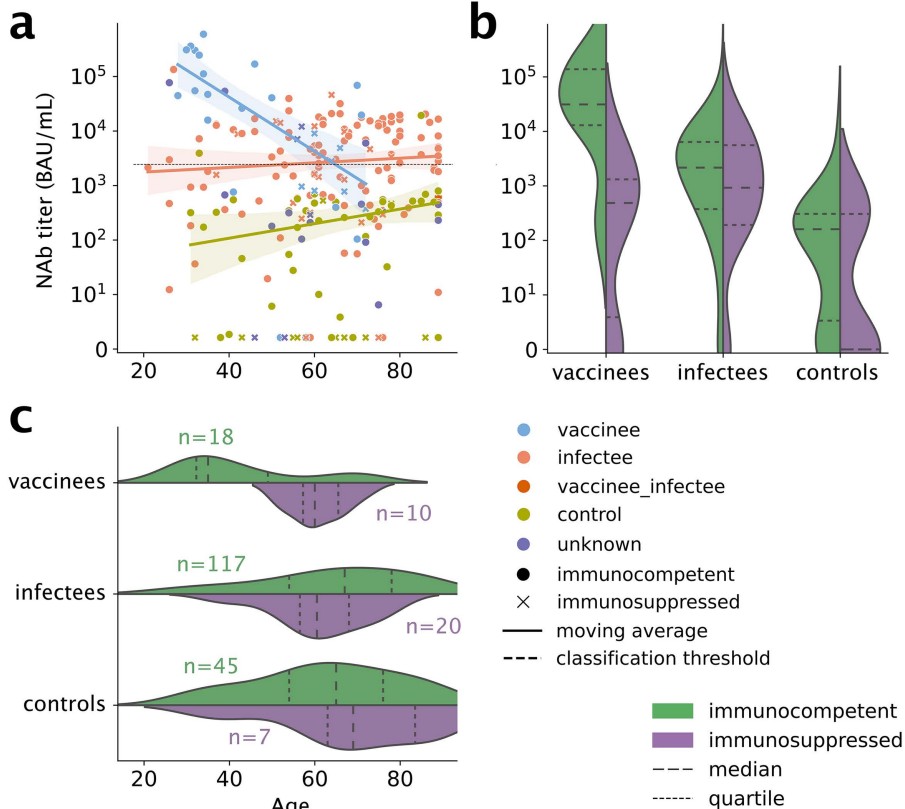

**Fig 1. Anti-SARS-CoV-2 ELISA trends and distributions by age for immunocompetent and immunosuppressed vaccinees, infectees, and controls.** NAbs target SARS-CoV-2 spike protein. (a) ELISA titers for each subject. Solid lines indicate regression fits; shaded areas indicate 95% confidence intervals. Dotted black line at ~$10^3$ indicates manufacturer's cutoff for positive vs. negative. Note strong negative trend with age in vaccinees (blue) but not infectees (salmon). Note mild positive trend with age in controls (olive), even as titers in this cohort remain below the cutoff for almost all individuals. (b) Distribution of titers in the three cohorts, split by immune status. (c) Distribution of ages in these cohorts, again split by immune status, with numbers of subjects in each sub-cohort.

residues ($p_c = 0.021$ vs. controls and $p_c = 0.0028$ vs. infectees) and fewer of the IGHV genes that contribute 4 residues (again $p_c = 0.021$ vs. controls and $p_c = 0.0028$ vs. infectees; Fig 2c and Table S4 in S1 File). The same pattern held for IgM-positive and IgM-negative (mainly IgG; hereafter "IgG") in controls and infectees. Meanwhile, the six IGHJ germline genes contribute 5 (IGH J4), 6 (IGH J3 and J5), 7 (IGH J1 and J2), or 10 (IGH J6) amino acids to the CDR3 (Fig 2d). We found that vaccinees used more J4 ($p_c = 9.4 \times 10^{-5}$ vs. controls and $p_c = 1.9 \times 10^{-6}$ vs. infectees) and fewer J3 & J5 ($p_c = 0.0002$ vs. controls and $p_c = 0.0021$ vs. infectees; Table S4 in S1 File). Thus, the *preference* of shorter IGH CDR3s after vaccination can at least partially be explained by selection for V and J genes that contribute fewer residues to the CDR3.

No such differences were observed in TCR CDR3s, which have a far narrower length distribution.

## Vaccination is associated with longer IGH CDR3s in non-productive joins

Next we sought to estimate the strength of selection for IGH CDR3s of different lengths in vaccinees, infectees, and controls. This can be done by comparing the length distribution of productive joins to the distribution in non-productive joins, i.e., those in which VDJ recombination occurs out of frame or produces stop codons. Because non-productive joins do not produce functional antibodies, the B cells that contain them cannot be selected for or against based on them. Nevertheless, the lengths of the CDR3 regions in non-productive joins can be measured. Thus, any differences in length between

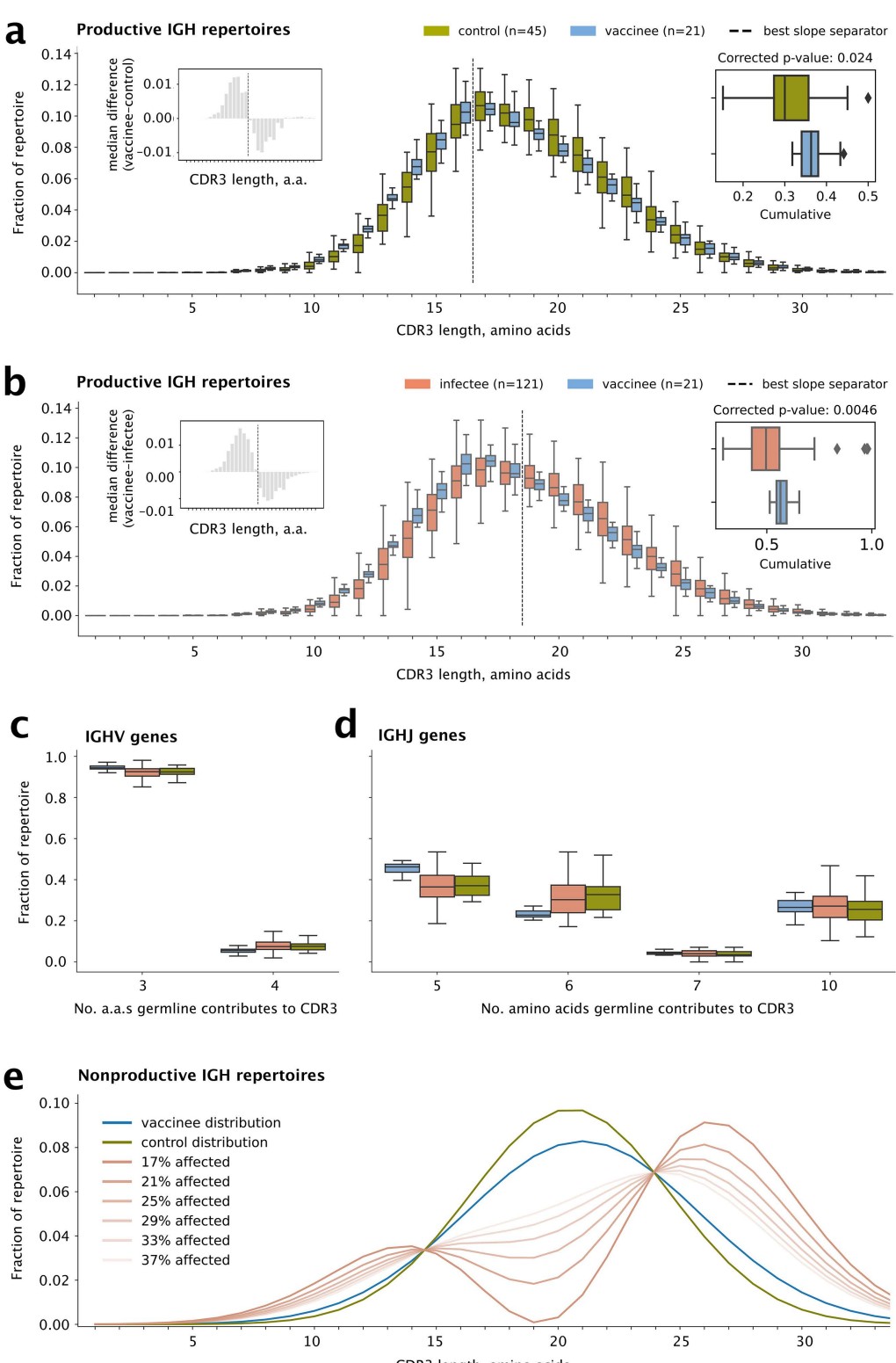

**Fig 2. IGH CDR3 length ddistributions.** (a) CDR3 length comparison plots for productive IGH repertoires of vaccinees vs. controls. Left inset: the median differences of frequencies at each length, showing that CDR3s of length 16 or shorter are more frequent in vaccinees, whereas CDR3s of length 17 or longer are less frequent. The pattern reverses at the dividing line between 16 and 17 amino acids (vertical dotted line). Right inset: total fraction

of the repertoire up to the dividing line. The p-value is obtained by applying Mann-Whitney U to the cumulatives followed by Holm-Bonferroni multiple-hypothesis correction. (b) The same for vaccinees vs. infectees, showing the same pattern but with a dividing line between 18 and 19 amino acids. (c)-(d) Frequencies of V and J genes grouped by the number of residues each gene contributes to the CDR3 according to germline (Fig S8 in S1 File). Note the only J gene that contributes 5 residues is IGHJ4. (e) Assuming the nonproductive IGH vaccinee repertoire (blue) is made up of a part that is unaffected by vaccination and therefore looks like the control repertoire (green) and a part that is affected by vaccination (salmon), this plot shows what the distribution of the affected part would have to look like so the two parts add up correctly, for different fractions affected (dark to light salmon lines). Estimated means for vaccinee and control distributions are shown. The smaller the affected portion, the more extreme the effect must be. The minimum possible effect size is that for which a CDR3 length for the affected portion is zero; any smaller, and a negative frequency at that CDR3 length would be required (negative frequencies are not possible). Boxes show the 25th and 75th percentiles. Horizontal lines show medians. Whiskers show the min and max unless there are outliers (diamonds).

non-productive joins and productive joins reflect selection on (some aspect of) the productive joins, for example by exposure to SARS-CoV-2 (in infectees) or vaccine contents (vaccinees).

Our null hypothesis was that the lengths of non-productive joins would be similar for vaccinees, infectees, and controls. Surprisingly, we found that CDR3s in non-productive joins differed across these three cohorts. In fact, we observed reverse relationships from the ones we saw in productive joins: CDR3s in non-productive joins were longer in vaccinees and infectees than in controls ($p_c$ = 0.039 and 0.0021, respectively). Vaccinees' non-productive CDR3s used the shortest J gene, J4, less often and the longest J, J6, more often than controls' ($p_c$ = 0.00011 and 0.022, respectively; Fig S5 in S1 File). Thus, in productive joins, selection for shorter CDR3s in vaccinees is even stronger than indicated from the comparison of productive joins in the previous section, because in vaccinees, recombination, which precedes selection, is biased toward longer CDR3s. Again, no such differences were observed in TCR CDR3s (Figs S6 and S7 in S1 File).

## Vaccination affects at least one-sixth of the pre-selection IGH repertoire

We next sought to better characterize this apparent effect of vaccine exposure on IGH recombination. The results in the previous section were regarding differences in subjects' entire IGH CDR3 repertoires. However, vaccine exposure is generally thought to affect only a portion of the repertoire. The rest of the repertoire, the unaffected portion, should be the same as a control's. Therefore, conceptually, each vaccinee's repertoire can be thought of as a weighted sum of two parts: a vaccine-responsive part and a control part. We asked what the minimum size of the vaccine-responsive part would have to be, in order to explain the difference in the length distribution of non-productive joins between vaccinees and controls.

To do this, we analyzed the differences between the mean IGH CDR3 length-distribution curves of vaccinees and controls. By calculating differences at each length, we generated the length distribution that the putative vaccine-responsive part would have to have, in order for the vaccinee curve to be a weighted sum of the control curve and the vaccine-responsive part, for a given size of the vaccine-responsive part (Fig 2e). Inevitably, there will be an inverse relationship between how different the length distribution of the vaccine-responsive part is, and its size. This fact sets a floor on the size of the vaccine-responsive part: any smaller, and the vaccine- responsive part would have to be so different that at least one of its lengths would have a negative frequency.

For example, 20-amino-acid-long CDR3s constituted an average of 9% of non-productive joins in controls but 8% in vaccinees. Considering just this length for the moment, if length-20 CDR3s constituted 7% in the vaccine-responsive part, then the vaccine-responsive part would have to be 50% of the repertoire, since 50% × 9% + 50% × 7% = 8%. If instead length-20 CDR3s constituted 3.5%, the vaccine-responsive part would only have to be 18%, since (100−18)% × 9% + 18% × 3.5% = 8%. In this example, the vaccine-responsive part could never be as small as 1%, since in that case length-20 CDR3s would have to have a negative frequency. By this approach, we found that the size of the vaccine-responsive part could be no smaller than 16%, or one-sixth, of the vaccinees' non-productive joins.

## Vaccinees and infectees with more SARS-CoV-2-specific TRBs have higher NAbs

Next, we tested whether TRB and IGH CDR3s that had been previously found to be associated with SARS-CoV-2 exposure, including by structural studies, were enriched among our vaccinee and infectee cohorts (see Methods). We obtained SARS-CoV-2-specific TCRs from CD4 and CD8 T cells from Nolan et al. [23] and obtained non-CD-restricted SARS-CoV-2- and non-SARS-CoV-2- specific TRBs and IGHs from CoV-AbDab, PDB, and VDJDb [40–42]. These comprised totals of 184,100 unique SARS-CoV-2-specific TRBs and 1,630 unique SARS-CoV-2-specific IGHs (Table S5 in S1 File).

We found a much higher proportion of SARS-CoV-2-specific TRB sequences than IGH sequences had exact matches in our samples: ≥ 12% vs. 0.1%, respectively, with the 0.1% representing just a single sequence (Table S5 in S1 File). The fraction of each repertoire that matched SARS-CoV-2-specific TRBs correlated positively with NAbs, as measured by ELISA titer, in infectees and vaccinees (Fig 3a-b, Table S6, and Fig S9 in S1 File). In infectees, for whom we had separate CD4-positive and CD4- negative (mainly CD8-positive; hereafter "CD8") TRB repertoires, the positive correlation was confined to CD4 repertoires. In contrast, no correlation was seen for controls. Likewise, no correlation was seen for TRBs that were not specific for SARS-CoV-2 in infectees, supporting the interpretation that this correlation is causal. Nevertheless, this correlation alone performed poorly as a classifier of who had high enough NAbs to be considered positive (per

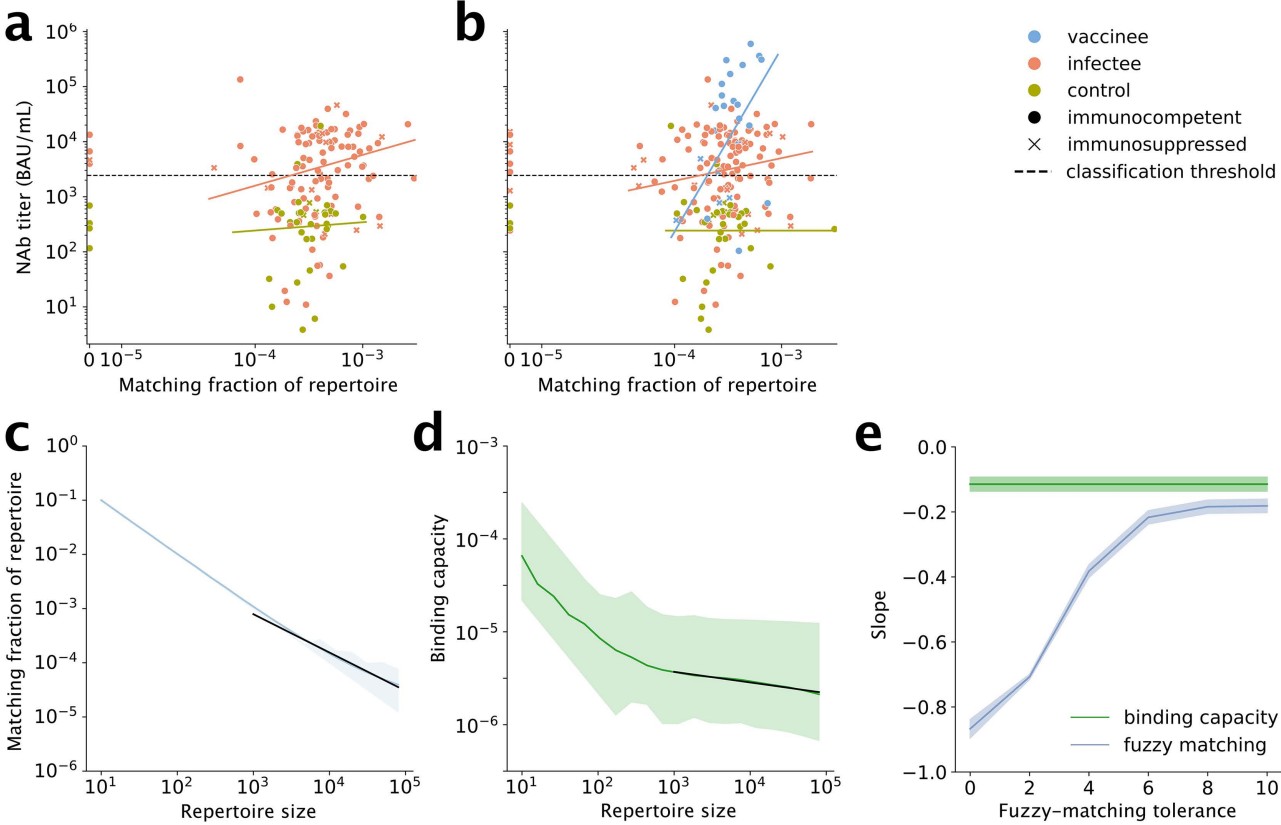

**Fig 3. SARS-CoV-2-Specific TRBs vs. NAb Titers.** (a)-(b) Fraction of TRB repertoires matching the SARS-CoV-2-specific CD4 TRB sequences obtained from Nolan et al.23 against SARS-CoV-2 NAb titer. Panel (a) shows repertoires from CD4 + T cells, which were available for infectees and controls but not vaccinees, while panel (b) shows repertoires from all T cells, which were available for all three cohorts. Theil-Sen regression fits (solid lines) show positive relationships for infectees and vaccinees but not controls. (c) The fraction of a repertoire that matches reference TRBs within a chosen tolerance (here, 2 amino-acid differences) depends strongly on the number of cells in the repertoire (i.e., repertoire size; Fig S10 in S1 File). (d) In contrast, binding capacity is much more robust. The slope of the dependency on size for repertoires above 1,000 cells are shown as black lines. (e) Slope as a function of fuzzy-binding tolerance, demonstrating binding capacity is more robust regardless of tolerance.

the ELISA test manufacturer), with an area under the receiver-operator characteristic curve (AUROC) of 0.55 (95%CI, 0.46–0.63). Notably, there was also a positive relationship between non-specific TRBs and NAbs in vaccinees, although the 95% CI on the regression slope only narrowly missed including zero (Table S6 in S1 File).

## Binding capacity outperforms fuzzy matching for measuring similarity

That subjects had almost no exact matches to SARS-CoV-2-specific IGH sequences did not exclude the possibility that they have sequences that are functionally similar to these reference sequences. The same possibility exists for TRBs. A standard method for finding similar sequences is using the Levenshtein distance (i.e., edit distance) (Fig S11 in S1 File). Sequences with a distance of less than or equal to a tolerance $t$ are considered similar (for example, sequences that differ by no more than $t = 1$ amino acid). This is known as "fuzzy matching" with tolerance $t$. (Note that exact matches are just fuzzy matches with tolerance 0.) Unfortunately, there is no consensus on what $t$ should be chosen. Also, the fraction of a repertoire that fuzzy-matches a set of references could depend on repertoire size because of the nature of sampling, potentially complicating the use of fuzzy matching.

To test this possibility, we subsampled 30 subjects' repertoires (10 controls, 10 infectees, and 10 vaccinees) and measured the fraction of the repertoire that fuzzy-matched SARS-CoV-2- specific CD4 TRBs at tolerances of 0, 2, 4, 6, 8, and 10 amino acids. We fit a linear mixed model grouped by subject for all repertoires with at least 1,000 sequences. We found the fraction of fuzzy matches depended strongly on repertoire size for all repertoire sizes measured (up to 1 million sequences), falling steeply and continuously throughout (Fig 3c). Thus, fuzzy matching is not a robust way to measure repertoires, at least in this study.

We therefore tested a recently proposed alternative method for finding similar sequences: measuring repertoires' binding capacity for the targets of reference sequences [43]. Binding capacity is the average similarity of a repertoire to one or more reference sequences, with similarity estimated according to a general model of the likelihood of a given sequence in the repertoire to bind the same antigen as a reference sequence. In contrast to fuzzy matching, we found the binding capacity remained robust for sample sizes above 1,000 sequences, with only minimal dependence on repertoire size (Fig 3d). Binding capacity was more robust to repertoire size than fuzzy matching at all tolerances tested (Fig 3e, Table S7 in S1 File; note that binding capacity does not require a choice of tolerance; it is independent of and therefore robust to tolerance; technically it is a nonlinear weighted mean across all tolerances). Thus, binding capacity provides a robust way to measure the fraction of these TRB repertoires that is similar to reference SARS-CoV-2-specific TRB sequences.

## Repertoire features predict levels of NAbs consistent with exposure comparably to clinical data

Finally, we compared how well above feature sets predicted exposure-level NAbs titers. To do so, we trained machine-learning models that used each of these feature sets. Because there were many reference SARS-CoV-2-specific TCR sequences to consider, each of which produces one exact-matching fraction, several fuzzy-matching fractions (one for each chosen tolerance), and one binding capacity measurement, there was a risk of overfitting (true whenever the number of features exceeds the number of datapoints). Therefore we first filtered out uninformative features.

To do this, we calculated exact/fuzzy matches and binding capacities for SARS-CoV-2 specific and non-specific sequences (from VDJDB) and measured their correlations to NAb titer. (Based on the results above, we only used repertoires with ≥ 1,000 sequences.) We used non-specific sequences as a null model and kept only SARS-CoV-2-specific sequences with correlations outside the middle 95% of the null model: specific sequences on the high end correlated more with NAb titer than was expected by chance, while specific sequences on the low end were correlated inversely with NAb titer to a larger degree than expected by chance (Fig 4a). Of 7,804 SARS-CoV-specific features with non-zero fractions or binding capacities, this process filtered out all but 323. To reduce redundancy and further reduce the number of features, we performed Principal Component Analysis on the results (the number of Principal Components to keep was

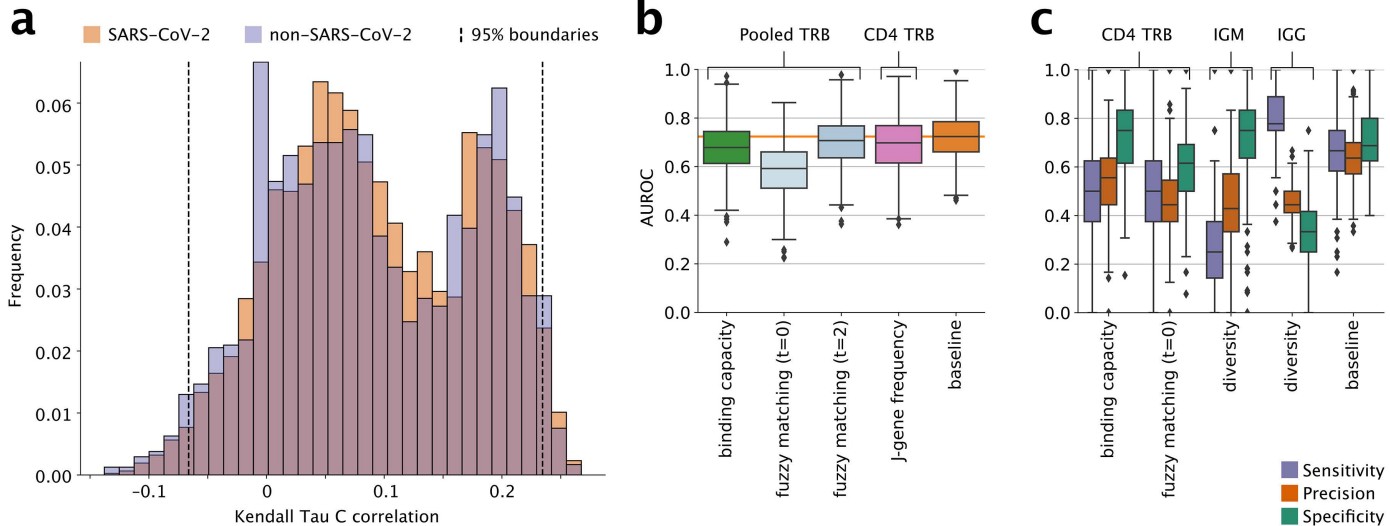

**Fig 4. Predicting positive NAbs.** (a) Feature selection mechanism used for binding capacity and fuzzy matching features on the binding capacity measurements of all TCR repertoires of size at least 1,000 using the SARS-CoV- 2-specific CD4 TCR sequences and non-SARS-CoV-2-specific TCR sequences obtained from VDJDB. Of the 7,804 SARS-CoV-2-specific features' correlations, 323 fall outside the selection boundaries set by the 95% boundaries of the correlations of non-SARS-CoV-2-specific features with NAb titer. (b)-(c) Machine learning performance results for a selected group of feature sets and cell types across all 700 replicates (100 repeats of 7-fold cross-validation). (a) shows areas under receiver operating curves and (b) breaks down the performances into sensitivity, precision, and specificity. The same plots for all feature sets and cell types can be found in Figs S12 and S13 in S1 File. Median values and interquartile ranges for all metrics are reported in Table S8 in S1 File. Boxes and whiskers as in Fig 2.

tunable and fit by each model). We did the same to reduce the number of V-gene features. To avoid data leakage, we performed this dimensionality reduction procedure on training data only.

We performed 700 replicate logistic-regression fits on each of the above feature sets and measured performance by AUROC (Fig 4b). As a comparator, we also fit 700 replicates on subjects' infection and vaccination status, which we reasoned would approximate the maximum possible performance that should be achievable on this dataset. As expected, this comparator resulted in the highest median AUROC of all the feature sets tested, at 0.72 (inter- quartile range across the replicates, 0.66–0.79; Fig 4b). Strong performance was also seen when training on fuzzy matches with tolerance 2 on all TRB sequences (AUROC 0.71; IQR, 0.64−77) and on TRJ frequencies for CDR4 TRB sequences (0.70; 0.62–0.77). Binding capacities on all TRB sequences showed similar performance to these two (0.68; 0.61–0.74), while exact matches on the same sequences showed poor performance (0.59; 051-0.66).

In sum, being infected and/or vaccinated—the gold-standard clinical model—lacked high predictive power for Nab titer, although binding capacities, fuzzy matches with a tolerance of 2, and TRJ frequencies on CD4 TRB sequences performed nearly as well and much better than exact matches.

To better understand the characteristics of different feature sets, we also calculated sensitivity, specificity, and precision for all replicates (see Fig 4c). The clinical feature set's performance metrics are relatively well balanced. In contrast, for most binding-capacity and fuzzy-matching feature sets, sensitivity and precision were low, making them less desirable for screening. Interestingly, features based on IgG diversity had the highest sensitivities while IgM diversities had the lowest sensitivities. The reverse was true for specificities, with IgM diversities having among the highest specificities and IgG diversities among the lowest. While it should be noted that repertoire diversity is not disease specific, these observations suggest that trends in diversity measurements taken for different repertoire subsets might give insights about exposure status in very different ways.

## Discussion

Detecting and defining signatures in repertoire sequence is challenging in part due to the large number of features that can contribute to a signature. These include high-level features such as CDR3 length and repertoire diversity, mid-level features such as frequencies of V and J gene usage and VJ combinations (D genes are harder to assign), and low-level features such as the frequencies of specific reference sequences. Repertoire diversity itself is actually a set of features, some of which incorporate sequence similarity, which furthermore can be defined in multiple ways [38,43]. Ideally the features above should be measured in both antibody and TCR repertoires, since they act cooperatively [44], and in cell subsets defined by isotype (for B cells) or CD4 vs. CD8 expression (for T cells). Thus, overall, the total number of features that can be used to detect and define signatures reaches into the hundreds of thousands.

As a result, statistical confidence requires large study sizes, which are challenging to obtain; methods that can avoid spurious associations, which are common in high-dimensional systems; appropriate controls, so that signatures are specific and not related to, e.g., general immune activation; and detailed clinical annotation, which we obtained from our electronic medical record (as detailed in Materials and Methods). Even with these design safeguards in place, the signature of exposure to a specific immunogen, such as SARS-CoV-2, may be broad or diffuse, with different individuals' repertoires reacting in different ways. And factors and features outside of repertoires may be important for determining exposure.

Given these considerations, our study was fairly large, with over 250 subjects, and involved sequencing IGH and TRD as well as TRB, to a median depth of over 105 cells/subject, made possible by ImmunoPETE's integrated library preparation.5 To focus analysis on SARS-CoV-2- specific signatures and patterns, controls in our study were not typical "healthy controls" but rather patients presenting for care who had sufficient concern for SARS-CoV-2 infection, and who were therefore tested, and were negative. At the time, hospital policy involved widespread testing with very sensitive tests (limit of detection, 100 copies of viral mRNA/mL), so we consider the probability of false negatives to be low. In addition, for infectees and controls, we separately analyzed IGM- and non-IGM (predominantly IGG+)-isotype antibodies and CD4+ and CD4- (i.e., mainly CD8+) T cells. Scheduling issues related to vaccine rollout prevented separate subset analysis for vaccinees, a limitation of the study. We also limited dimensionality, thereby increasing statistical confidence, by filtering for features that correlate with the outcome measure of NAb titer. And instead of simply combining all features into a single model, we compared models with different feature sets to tease apart where signals might lie. Finally, we compared these to the simplest model we could think of, made up of readily available clinical information: whether or not a person was infected and/or vaccinated, to test how repertoire data compares (and what, if anything, it could add). To our knowledge this is the largest such study, and possibly the first. It led to the discovery of several previously unreported patterns across multiple feature sets, for both IGH and TRB, as well as in multiple subtypes of B and T cells, that merit discussion.

First, the pattern in IGH CDR3 lengths in vaccinees was curious for several reasons. First, it involved a change in non-productive joins (which in our reading of the literature are usually treated as a baseline and not compared between cohorts, as we did). This was unexpected because B cells are selected for survival based on expressed B-cell receptors, and non- productive joins are not expressed. Our finding seems to indicate selection independent of expression (non-productive joins are not expressed). Second, this is a much larger effect than would be expected from an antigen-specific adaptive immune response. Immunogen-specific B cells rarely exceed low-single-digit percentages of the repertoire. Yet the effect we found appears to involve at least one-sixth (~17%) of the repertoire. Third, the direction of the length change in non-productive joins is opposite that of productive joins: Non-productive CDR3s are longer in vaccinees than in controls and infectees, but productive CDR3s are shorter. And fourth, while other patterns we found were fairly similar between vaccinees and infectees, this CDR3 length effect appears confined to vaccinees.

We conclude that vaccination may have some undescribed effect on the V-D-J recombination machinery, biasing recombination toward use of IGHJ genes (and secondarily IGHV genes) that result in longer CDR3s. This effect would have to be due to some difference between the vaccine and natural infection, or else it would have been seen in infectees. If our interpretation is correct, it would mean the effect of selection for shorter CDR3s in productive joins is quite strong,

because there are fewer short joins from which to select. In any event, both vaccination (in nonproductive and productive joins) and infection (in productive joins) affect a larger proportion of IGH repertoires than is typically considered "specific."

Second, binding capacity was shown to have essentially the same predictive power as the best- performing version of fuzzy matching. Recall that both fuzzy matching and binding capacity measure the size of groups of similar antibodies or TCRs. Here they were applied by taking a reference sequence, for example a sequence previously reported in the literature to be associated with SARS-CoV-2 (a "SARS-CoV-2-specific sequence") and ask what fraction of a given subject's repertoire was similar to that index sequence. The methods differ in how they view similarity. Fuzzy matching requires choice of tolerance: above a set number of amino-acid mismatches, a query sequence is considered different to the index sequence. If the tolerance is 2, a query with 3 mismatches is considered just as different from the index sequence as a query with 20 mismatches.

Binding capacity has neither problem. It is based on the measured relationship between number of mismatches and change in dissociation constant (Kd), i.e., binding similarity [43]. This empirical data essentially substitutes for having to choose a tolerance. In addition, binding capacity is continuous: a query with 3 mismatches is more similar to the index than a query with 20 mismatches. Consequently, binding capacity can detect the potential presence of a large group of sequences with low similarity, which collectively might play as important a role as a small group of high-similarity sequences (or in the limit, the presence of the index sequence as a high-frequency clone). The magnitude of the CDR3 length effect supports the importance of being able to detect such diffuse/weak signals. We showed that different tolerances had different ability to predict NAb titer. To us there is no obvious reason that tolerance of 2 should outperform, e.g., a tolerance of 10. Possibly which tolerance is best may differ by exposure. That binding capacity performs comparably to the best- performing tolerance supports its utility for immune-repertoire analysis.

We note that in principle, diffuse effects could reflect superantigen-type activity [45]. However, to our knowledge, clinical evidence for a superantigen in SARS-CoV-2 is scant and has focused on T cells. Although proposed to explain the rare sequela of multisystem inflammatory syndrome in children [46], wet-lab investigation has not demonstrated superantigen activity in SARS-CoV-2 [47].

This study has several limitations. First, we were unable to sort vaccinee samples to obtain separate IgM vs. IgG B-cell repertoires and CD4 vs. CD8 T-cell repertoires due to exigencies at the height of the pandemic. Different subtypes may follow different (even opposite) trends, as did the sensitivities and specificities of classifiers trained on IGG and IGM diversities. Any such patterns in vaccinees were beyond our ability to measure. Second, we used concentrations of SARS-CoV-2 anti-spike NAbs as our proxy of protection. Signals may be present that do not correlate with antibodies binding this particular immunogen. For example, a signal might be seen in T cells or antibodies that bind other SARS-CoV-2 proteins, which we are unable to evaluate given NAbs as a readout. Third, although the sequence data in this study was quantitative, it contained only single-chain, not paired-chain data. Fourth, the ability to define signatures is limited by uncertainty about the specificity of reference sequences. Much effort is being put into methods that predict receptor-antigen binding, but a unified, accepted, and feasible approach to identifying all sequences that bind a given immunogen has yet to be established. Fifth, the quality of binding capacity measurements is limited by the current measure of binding similarity being based on mean behavior [43]; this is expected to improve with additional data and advances in protein structure prediction. And sixth, we did not have the benefit of HLA typing for these samples, which could be valuable because of its influence especially on the TCR repertoire.

It will be valuable to see the methodology presented here, with its many steps taken to maximize robustness and avoid statistical artifacts, applied to additional datasets. This will give additional evidence of how well these results and this approach generalize for SARS-CoV-2 in general, for immune responses to variants of the virus, and for other pathogens and immunogens. A careful statistical approach applied to multiple, functional features, measured on unbiased repertoire sequence from TCR and BCR subsets from large cohorts, is, in our opinion, the best way to decipher the rich information that the adaptive immunome encodes.

                                                                                          

## Materials and methods

### Study subjects

The subjects in this study were patients seeking clinical care at the Beth Israel Deaconess Medical Center (BIDMC), a 743-bed tertiary care medical center in Boston, MA, USA. BIDMC serves a large and diverse population in and around eastern Massachusetts, USA, centered on the Boston metropolitan area. The recruitment period was November 12, 2020 to April 12, 2021. Consent was waived under IRB approval (see below).

### Institutional review board (IRB) approval

All work was carried out in accordance with BIDMC's Institutional Review Board protocols 2020P000634, 2021P000109 and 2020P000361.

### Cohort assignment

All subjects from whom samples were obtained received RT-qPCR tests performed on two Abbott Molecular platforms: m2000 and Alinity m (Abbott Molecular, Des Plaines, IL, U.S.A.). These detect identical SARS-CoV-2 N and RdRp gene targets and are extremely sensitive for SARS-CoV-2 infection, with limit of detection of 100 copies/mL [48–50]. Infectees had a positive result at the time of sample acquisition. Controls were tested, but negative. COVID-19 test and vaccination information were obtained using SQL queries from BIDMC's clinical data repository and via a dedicated REDCap data-base set up to facilitate research involving vaccinees [51].

Using these records, subjects were considered infectees if there was a record of a positive COVID-19 test result dated before or on the sample collection date and non-infected otherwise. If no medical record number was available for a subject, their infection status was considered unknown. Subjects were considered vaccinees if vaccination prior to or on the day of sample collection was indicated as the appropriate procedure code in the clinical data repository, recorded in REDCap, or identified from Massachusetts' state Immunization Information System. The entire vaccinated cohort had received mRNA-based vaccines, with 80% receiving Pfizer Comirnaty and the rest receiving Moderna Spikevax. Samples were collected from the vaccinee cohort 4–84 days (mean = 44.3 days, standard deviation = 15.3 days) after administration of the initial vaccine dose.

Subjects were considered non-vaccinated if the sample collection date preceded 12/15/2020 (the date of the first administered COVID-19 vaccine); if there was record of vaccination after sample collection that was annotated as the first dose; if there were two vaccinations after sample collection where the second was annotated as the second dose; or if there were two vaccinations after sample collection within 42 days of each other (consistent with being the primary series). Subjects that did not satisfy vaccinee or non-vaccinated criteria were considered to have unknown vaccination status. Subjects were annotated as unexposed controls if they were non-infected and non-vaccinated. Subjects whose vaccination status was unknown or whose infection status was unknown and were neither vaccinees nor infectees were considered to have an "unknown" SARS-CoV-2 exposure status.

### Clinical annotations

**Immunosuppression.** Subjects were labelled either "immunosuppressed" or "immunocompetent." Subjects were designated immunosuppressed if at least one of the following criteria was met:

- the most recent CD4+ cell count was less than 100 cells/μl;

- there was a diagnosis of lymphoma or leukemia associated with a healthcare encounter (visit, admission, or phone call) either before or within 60 days after sample collection;

- or

- the subject was prescribed any of the following medications on an ongoing basis prior to sample collection and with enough refills to include up to 30 days after: abatacept, adalimumab, anakinra, azathioprine, basiliximab, budesonide, certolizumab, cyclosporine, daclizumab, dexamethasone, everolimus, etanercept, golimumab, infliximab, ixekizumab, leflunomide, lenalidomide, methotrexate, mycophenolate, natalizumab, pomalidomide, prednisone, rituximab, secukinumab, sirolimus, tacrolimus, tocilizumab, tofacitinib, ustekinumab, and vedolizumab.

If none of these criteria were met, subjects were considered immunocompetent.

**Demographics.** If a subject had a COVID test, the sex and date of birth were read from the corresponding record. Otherwise, we read sex and date of birth from other records of lab specimens, the electronic health record (EHR), or the project's REDCap database (always in structured fields, not using natural-language processing). Self-reported race was read from the EHR.

**Risk factors.** A semi-automated review of EHRs for ICD-10 diagnosis codes and related entries was used to identify subjects having any of the medical conditions highlighted by the CDC as increasing risk of severe illness from COVID-19 [52]. Where feasible, the list of ICD-10 codes indicative of each comorbidity was taken from the Elixhauser Comorbidity Software Refined for ICD-10-CM [53], version v2022.1, developed for the Healthcare Cost and Utilization Project (HCUP), which is based on the work of Elixhauser et al. [54] In addition to these, another widely used set of comorbidity measures is the Charlson Comorbidity Index [55]. For comorbidities not defined in the HCUP software, the lists of ICD-10 codes defined by this study56 were used where possible. Comorbidities that were not codified in either resource were identified, where possible, using ICD-10 codes or other automated chart queries, detailed as follows:

- Cancer: identified using ICD-10 codes in the HCUP software for "Leukemia," "Lymphoma," "Metastatic cancer," or "Solid tumor without metastasis, malignant."

- Chronic Kidney Disease: identified using ICD-10 codes in the HCUP software for "Renal failure, moderate," and "Renal failure, severe."

- Chronic Liver Disease: identified using ICD-10 codes in the HCUP software for "Liver disease, mild," and "Liver disease, moderate to severe."

- Chronic Lung Disease: The CDC website stipulates that asthma is of concern "if it's moderate to severe," implying mild asthma is not of concern. The HCUP software includes codes for all degrees of severity of asthma in the definition of "Chronic pulmonary disease." Thus, chronic lunch disease was identified using ICD-10 codes in the HCUP software for "Chronic pulmonary disease," excluding any ICD-10 codes beginning with J452 or J453 (mild intermittent or mild persistent asthma, respectively).

- Cystic Fibrosis: Identified by any ICD-10 code beginning with E84.

- Dementia or other neurological condition: identified using ICD-10 codes in the HCUP software for "Dementia," "Neurological disorders affecting movement," "Seizures and epilepsy," and "Other neurological disorders."

- Diabetes: identified using ICD-10 codes in the HCUP software for "Diabetes with chronic complications" and "Diabetes without chronic complications."

- Disabilities: identified using ICD-10 codes in the HCUP software for "Paralysis" plus any ICD-10 code beginning with Q (birth defects and chromosomal abnormalities). Note that this omits many, possibly most, forms of disabilities, including non-congenital blindness and deafness, cognitive impairments not due to chromosomal abnormalities, autism spectrum disorders of unknown etiology, etc., but these are of dubious connection to COVID-19.

- Heart conditions: identified using ICD-10 codes in the HCUP software for "Heart failure," the ICD-10 codes listed in the referenced study[50] for "Myocardial Infarction," and/or any ICD-10 code starting with any of these prefixes: A1884, A3282,

A3681, A381, A395, A5203, B2682, B332, B376, B5881, C452, D8685, G130, G712, G713, G720, G721, G722, G7249, G7281, G7289, G729, G737, I01, I02, I05, I06, I07, I08, I09, I11, I13, I20, I23, I24, I25, I3, I4, I5, I70, I9713, J1082, J1182, O101, OO2912, O903, Q2, R570, S26, T82, and Z95.

- HIV: identified using ICD-10 codes in the HCUP software for "Acquired immune deficiency syndrome."

- Mental health conditions: identified using ICD-10 codes in the HCUP software for "Depression" and "Psychoses." Note that this may omit many other forms of mental illness, such as obsessive-compulsive disorder, post-traumatic stress syndrome, borderline personality disorder, etc. Note that there is overlap between conditions considered mental health conditions and those considered disabilities (such as autism spectrum disorders) as well as between mental health conditions and other medical conditions (such as substance abuse disorders).

- Overweight or obese: Subjects were considered to be overweight or obese if their BMI was ≥ 25. If multiple BMI or height-and-weight values were recorded in the database over time for a given subject, the value(s) used were those closest in time to the date of sample collection.

- Pregnancy or recent pregnancy: Electronic medical records of all female subjects under the age of 69 were searched for: ICD-10 codes starting with Z3A and records of hospital admissions which include a baby delivery time. The timespans of the pregnancy and puerperium periods were estimated from either type of record. In the case of ICD-10 codes starting with Z3A, the final digits of the ICD-10 code encode weeks of gestation at the time of the encounter, from which a start and end date of the pregnancy can be estimated. If only a delivery date is known, the pregnancy is estimated to have begun 40 weeks earlier, unless "PRETERM" is found in the free-text diagnosis. Subjects were marked as "pregnancy or recent pregnancy" only if their COVID-19 test date fell between the estimated start date of the pregnancy and 42 days after the estimated end date (to allow for post-term pregnancy). Where there was no COVID test date, the date of the blood sample collection was used.

- Sickle cell or Thalassemia: Identified by any ICD-10 code beginning with D56 or D57.

- Smoking, current or former: Electronic medical records were searched for any non-zero "Tobacco pack years," and for a free-text description of their tobacco usage including the text "current smoker," "former," "every day," "some days," "light," "heavy," "less than 10," "10+," "yes," or "counseling provided."

- Solid organ or blood stem cell transplant: Identified by any ICD-10 code beginning with Z94.

- Stroke or cerebrovascular disease: identified using ICD-10 codes in the HCUP software for "Cerebrovascular disease," which includes ICD-10 codes for both CBVD POA and CBVD SQLA.

- Substance abuse: identified using ICD-10 codes in the HCUP software for "Drug abuse" and for "Alcohol abuse."

- Tuberculosis: Identified by any ICD-10 code beginning with A15.

### Sample collection, cell separation, and DNA extraction (Figs S14-S15 in S1 File)

2mL aliquots were taken from EDTA-anticoagulated venous blood collected in the course of standard clinical care (via "purple-top" tubes; Becton, Dickinson, Franklin Park, NJ). Tubes were stored at 4°C between collection and processing, never more than 12 hours. Each aliquot was mixed 1:1 dilution in phosphate-buffered saline (PBS) and centrifuged over Ficoll-Paque-plus (Cytiva, Marlborough) to obtain peripheral blood mononuclear cells (PBMCs). Plasma was collected and stored at 80°C. PBMCs were washed with PBS and resuspended in a sorting buffer of PBS, 1% bovine serum albumin (BSA), and 0.01% sodium azide.

Magnetically-labeled anti-CD4 and anti-IgM microbeads (Miltenyi, Bergisch Gladbach) were used to label and column-separate for infectee and control samples; vaccinee samples cells were not separated. This process divided the

samples into CD4-positive T cells and IgM-positive B cells in one fraction, and CD4-negative T cells and IgM-negative B cells (principally CD8- positive T cells and IgG-positive B cells) in the other fraction. DNA was isolated for each fraction using EZ1&2 DNA Blood 350 μL kits (Qiagen, Hilden) and the EZ1 Advanced XL automated system (Qiagen, Hilden). DNA concentration was assessed via Nanodrop (Thermo Fisher, Waltham).

## Sequencing library preparation

AIRRseq libraries were generated using the immunoPETE method as described [5]. ImmunoPETE is a two-step primer extension based targeted gene enrichment assay designed to specifically target and quantitatively amplify recombined human TRB, TRD, and IGH from genomic DNA simultaneously. Briefly, V gene-based primers containing unique molecular identifiers (UMI) as well as universal PCR amplification handles were annealed to the chromosomal VDJ rearranged loci. The first primer extension products, spanning the VDJ rearrangement, were purified from any remaining oligos by a combination of beads (KAPA HyperPure, Roche) and enzymatic treatment with Thermolabile Exonuclease I (New England Biolabs). A second primer extension and amplification master mix containing a pool of J-gene oligos and an Illumina i7 primer generated VDJ amplicons after 10 cycles of target amplification. Illumina sequencing library amplification was performed using the i7/i5 primer pairs with dual sample indexes. All primer extensions and amplifications were performed using the KAPA Long Range HotStart Ready Mix (Roche). The resulting libraries underwent purification using KAPA HyperPure beads (Roche), followed by quantification with the Qubit dsDNA HS Assay kit (Thermo Fisher) and fragment analysis (Agilent TapeStation). Individual sample libraries were pooled in equal mass. A final round of quantification and fragment analysis was then performed. Finally, libraries were sequenced using the Illumina NextSeq 500/550 High Output Kit v2.5 (300 cycles).

## Sequencing and bioinformatics

ImmunoPETE sequencing libraries were analyzed using the Roche in-house bioinformatics pipeline, Daedalus (https://github.com/bioinform/Daedalus). After quality filtering of reads and trimming off primers, the pipeline identified V and J genes using a Smith-Waterman alignment approach (https://github.com/pgngp/swift) against an in-house curated V and J gene database. Original V and J gene data and sequences were sourced from HGNC (https://www.genenames.org/) and ENSEMBL (https://ensemblgenomes.org/). CDR3 sequences were identified for all V-J pairs, capturing both functional (functional V/J gene AND coding CDR3) and non-functional (annotated non-functional or pseudogene V/J gene in the database OR stop codon/frameshift in CDR3) rearrangements. Sequences are deduplicated by clustering UMI and CDR3 sequences to identify UMI families. Consensus sequences were derived for the CDR3 and UMI segments of each UMI family, suppressing sequencing and PCR errors, and identifying CDR3 rearrangements at single molecule resolution. High quality CDR3 rearrangements were further analyzed for cell counting, clonal diversity, and other calculations. Terms used are listed alphabetically and defined as follows:

- Cell count: the total number of functional IGH, TRD, and TRB rearrangements in a sample

- Cell type percentages: the total number of functional rearrangements from each heavy chain divided by the total cell count × 100

- CDR3 clone: BCR or TCR sequences from the same individual with matching V gene,

- CDR3 amino acid sequence (CDR3-AA), and J gene assignment arising from two or more UMI families

- CDR3 clonal type: BCR or TCR sequences from multiple UMI families from multiple individuals with matching V gene, CDR3-AA, and J gene assignment

- Clone count: total number of UMI families from the same individual with the same V gene, CDR3-AA, and J gene

 

- UMI family: a set of reads that have been clustered together based on the similarities of the 9-nt UMI sequence and the CDR3-nt region

Both UMI and CDR3 sequences are clustered based on a Levenshtein edit distance of 1, capturing likely PCR and sequencing errors. A UMI family represents a single captured DNA molecular fragment from the immunoPETE reaction.

## NAbs ELISA titers

The SARS-CoV-2 Surrogate Virus Neutralization Test Kit (GenScript, L00847-A) is designed to quantify the circulating antibodies capacity of blocking the interaction between the receptor binding domain of SARS-CoV-2 spike glycoprotein and the ACE2 human cell surface receptor. The kit was used according to the manufacturer's instructions as follows. A standard curve was generated using a serial dilution of the standard (GenScript, A02087-20) with a dilution factor of 1:2. Each subject's serum sample was mixed with sample dilution buffer (1:10) and horseradish peroxidase-conjugated recombinant SARS-CoV-2 receptor-binding domain (HRP-RBD). The mixture was incubated at 37°C for 30 min to allow the circulating NAbs to bind to HRP-RBD. The mixture was then added to an ACE2 protein-coated plate and incubated for an additional 15 min at 37°C. Unbound HRP-RBD and HRP-RBD bound to non-neutralizing antibodies were bound to the plate while circulating neutralization antibody HRP-RBD complexes remained in the supernatant for subsequent wash steps. After washing, tetramethylbenzidine solution was added, followed by a stop solution to quench the reaction, turning wells yellow. The plate was read immediately at 450 nm in a microtiter plate reader. Statistical analysis was performed with GraphPad Prism using a 4PL model for linear regression. Results were reported by interpolating the OD450 values to the standard curve values.

Results are reported in the WHO standard units (BAU/mL), which can be converted to the manufacturer's U/mL by dividing numbers by a factor of 1.626 (per the manufacturer).

## pymmunomics

Code used for the analyses was written up as a python package and made publicly available on github (https://www.github.com/ArnaoutLab/pymmunomics). Reference is made in the following sections wherever that is the case.

## Validation of pooled sequences from sorted subsets vs. unsorted sequences

Additional test samples were split into two replicates. One replicate was sorted into IgM/CD4 and IgG/CD8 fractions as above. The other replicate was unsorted. Each of the IgM/CD4, IgG/CD8, and unsorted fractions had libraries prepared as above and were sequenced as above. V- and J-gene frequencies from the IgM/CD4 and IgG/CD8 fractions and from the unsorted fraction were measured, and the frequencies from the IgM/CD4 and IgG/CD8 fractions were averaged together. These averages were compared to the frequencies in the unsorted fraction, and were shown to be indistinguishable by t-test (p-values ranging from 0.19 to 0.95 for comparisons of the frequencies of each gene), demonstrating that IgM/CD4 + IgG/CD8 is statistically indistinguishable from unsorted, supporting comparisons between infectees (sorted) and vaccinees (unsorted) and between controls (sorted) and vaccinees. Dependence of antibody concentrations on age, immunocompetence, and SARS-CoV-2 exposure.

Univariate and bivariate exploratory plots suggested zero antibody concentration to be a special category. Therefore, we first modeled the ability to produce zero vs. non-zero amounts of antibody using logistic regression. We then performed linear regression to model the $\log_{10}$- transformed concentration of the nonzero values on our set of covariates. In both cases, we started with a full model incorporating age, immunocompetence status, cohort, and all of their two-way and three-way interactions. Starting with the interaction terms and then proceeding to the main effects, we sequentially eliminated covariates that were not significant at α = 0.05. This did not change the regression coefficients of any of the significant terms by >20% (i.e., were not confounders). Finally, we confirmed that the best model had lower AIC (logistic regression) or higher adjusted $R^2$ (linear regression) compared with the alternative models.

## CDR3 length analysis

For calculating lengths, the CDR3 region was defined to include the preceding Cysteine anchor residue and the terminating Tryptophan (for IGH) or Phenylalanine (for TCR) anchor residue. CDR3 length frequencies for each available functional and non-functional pooled IGH, TRB, TRD, and subtyped IGG, IGM, CD4 TRB/D, CD8 TCB/D repertoire of immunocompetent subjects were calculated using the pymmunomics python package (above). Since vaccinee samples were not sorted into subtypes, pooled repertoire CDR3 length frequency distributions were used to compare vaccinees to controls and infectees. CD4+/IGM+ and CD8/IGG repertoire CDR3 length frequency distributions were compared independently between controls to infectees.

To compare CDR3 length distributions between cohorts without simplifying them down to their mean or median distribution, which ignores variance within groups, we chose a threshold CDR3 length $\ell$ and compared the cumulative frequencies of sequences on each side of that length using a two-tailed Mann-Whitney-U test. The threshold length was determined by estimating the difference of length frequencies between cohorts for each CDR3 length. These estimates were calculated by taking the median difference in frequency between members of one cohort and members of the other. The dividing line is then placed between the lengths $\ell$ and $\ell + 1$, where $\ell$ is the CDR3 length that maximizes the magnitudes of the total areas under the curve of estimated frequency differences to the left and right of the line, i.e., the best dividing line between patterns:

$$\left| \sum_{\ell' < \ell} d_{\ell'} \right| + \left| \sum_{\ell' > \ell} d_{\ell'} \right|$$

Here $d_{\ell'}$ denotes the estimated difference of frequencies of CDR3s of length $\ell'$ between the two cohorts. Note that the absolute values are taken after summing group differences on one side of the dividing line (making positive and negative differences cancel each other out before taking the absolute value), favoring a dividing line that splits the median differences into large same-signed runs. P-values were corrected for multiple hypotheses via the Holm-Bonferroni method (Table S3 in S1 File).

To identify trends among lengths of V and J genes, V and J genes (from IMGT) of the relevant cell types (IGH for the functional pooled IGH comparisons and pooled IGH, IGG, and IGM for the non-functional comparisons) which had a corrected p-value below 0.05 were grouped into the number of residues that fall into the CDR3 region. Usage frequencies of V- and J-gene groups were compared between cohorts using two-tailed Mann-Whitney-U and a second correction round was conducted to correct all original p-values of the CDR3 length comparisons at the same time as the p-values obtained from the follow-up tests.

## Sets of known SARS-CoV-2 binders and binders to other pathogens

MIRA-identified SARS-CoV-2 specific T-cell receptor sequences [23] were downloaded from https://clients.adaptivebiotech.com/pub/covid-2020 on April 19, 2021.

Query B-cell and T-cell receptor sequences (CDR3) of cells known to bind to SARS-CoV-2 were downloaded from CoVAbDab, PDB, and VDJDB. The CovAbDab sequences were downloaded on April 20, 2022 and consists of all SARS-CoV-2-WT-neutralizing human antibodies with CDRH3 sequence listed in the database at the time and added since May 04, 2020. PDB sequences were download on May 03, 2022 searching for all structures of source organism Homo sapiens, containing in the title one of "antibody" or "Fab," and one of "CMV," "cytomegalovirus," "DENV" (i.e., dengue virus), "dengue," "EBV," "Epstein-Barr," "hepatitis," "HIV," "human immunodeficiency virus," "influenza," "SARS-CoV-2," or "tetanus." The resulting entries were filtered for sequences in which a CDRH3 sequence of length at least 6 and at most 40 could be detected using in-house Python code. For each sequence, the name of the binding target was extracted from the structure title. VDJDB sequences were also downloaded on April 20, 2022 to obtain human TRB sequences with CDR3 and J-gene specified that bind to their listed target with a non-zero score.

To conform with the gene database used for V- and J-gene assignment of repertoire sequences (see Sequencing and bioinformatics), the same gene sequences were aligned (blastp and blastp-short for V genes and J genes, respectively; BLAST+ v2.12.0) to the sequences from PDB and CoVAbDab, setting the max target seqs parameter to 10,000—a number much larger than the total number of genes in the query to avoid missing the best matching genes [17]. V-gene matches with query coverage less than 30% or percent identity less than 40% and J-gene matches with query coverage less than 50% or percent identity less than 40% were filtered out. From the remainder, the best V- and J-gene matches according to percent identity and gene sequence coverage (lexicographically) were assigned to each query sequence. Data downloaded from VDJDB contained sequence only for the CDR3 region, so the V, and J-gene annotation provided by the database was used (as opposed to using, e.g., BLAST).

To calculate the fractions of query sequences sets matching subject repertoire sequences and the fractions of subject TRB repertoires matching query TRB sequences sets, a pair of sequences is considered to match if their V gene, J gene, and CDR3 sequence are identical.

## Binding-capacity measurements

Binding capacities to the MIRA-identified HLA class II T-cell sequences were measured for all subject pooled (CD4 + CD8), and CD4 TRB repertoires, wherever possible. The binding capacity of a repertoire R to a clone c is defined as:

$$\tau(c; R) = \sum_{c' \in R} p(c') \cdot s(c, c')$$

where $p(c')$ denotes the frequency of clone $c'$ in repertoire $R$ and $s$ is the binding similarity between sequences. Here, s as previously described [43], which accounted only for the relationship between Levenshtein distance of CDR3s and the predicted difference in strength of their binding to the same target(s) (in terms of relative $K_d$), was constrained as follows to require matching V and/or J genes:

$$s(c, c') = \begin{cases} 0.3^{Lev(c,c')} & \text{if V and J genes match} \\ 0 & \text{otherwise} \end{cases}$$

Here, $Lev(c, c')$ is the Levenshtein distance between the CDR3 amino acid sequences of sequences $c$ and $c'$. The pymmunomics Python package was used to calculate similarity matrices and binding capacities.

## Fuzzy query sequence matching

Fuzzy sequence matching measurements for each pooled CD4 + CD8 and each CD4-only TRB subject repertoire to the MIRA-identified HLA class II query sequences were tabulated from the similarity matrices that are calculated as part of determining binding capacities. For each query sequence and each subject repertoire, we measured the fraction of repertoire sequences for with the same V and J genes as the query sequence, and whose CDR3 sequence was within Levenshtein distances 0–10 of the query's CDR3. Note that exact matching is equivalent to fuzzy matching with a Levenshtein distance of 0.

## Binding-capacity and fuzzy-matching robustness experiments

To compare robustness to variations in repertoire size of binding capacity and fuzzy matching features, we conducted subsampling experiments. We randomly chose 10 subjects from each of the vaccinee, infectee, and control cohorts that had a pooled TRB repertoire size of at least 80,000 cells, i.e., 80,000 distinct corrected UMIs. (This size was chosen in order to guarantee at least 10 subjects from the control cohort to choose from.) Each of these repertoires was sampled down

to 20 different subsample sizes chosen to be equidistantly spaced between 10 and 80,000 at log-scale. For each subsample, we calculated binding capacities as well as fraction of fuzzy matches for fuzzy-match tolerances 0, 2, 4, 6, 8, and 10 amino acids to CD4 TRB reference sequences from MIRA. The slopes and their surrounding 95% confidence intervals were obtained by fitting a linear mixed model that groups the data by subject.

## Feature selection

Preferring the use of domain knowledge over generic feature selection mechanisms for selecting from the high-dimensional query sequence matching features (binding capacity and fuzzy matching), a custom feature selection method is developed and implemented in the python package pymmunomics. For this mechanism we use binding capacity and fuzzy matching measurements to sequence specific to pathogens other than SARS-CoV-2 ("SARS-CoV-2 non-specific sequences") as a null distribution to which to compare the measurements for MIRA-identified SARS-CoV-2-specific sequences. We calculated the (Stuart-)Kendall Tau-c correlation coefficient between each feature's measurement and NAb titer. For each feature group (binding capacity, fuzzy matching with tolerances 0, 1, …, 10, etc.), the correlation coefficients of measurements for non-SARS-CoV-2 specific sequences form the null distribution and correlation coefficients of SARS-CoV-2 specific features below the 2.5th and above the 97.5th percentile are selected (cumulatively, the most correlated and anti-correlated 5%).

Following the same idea, V-gene frequencies were also selected from among the 54 total possibilities (one for each V gene). Here, V-gene frequencies in non-functional repertoires were taken as the null distribution against which to compare functional repertoires' V-gene frequencies, since non-functional sequences do not undergo SARS-CoV-2 specific clonal expansion. Since the functional and non-functional frequencies can be viewed as paired measurements, the distribution of differences between their correlation coefficients was calculated, and the most correlated and anti-correlated 5% (as defined above) were selected as features.

## Machine learning to classify subjects with a protective NAb titer

Machine learning classifiers of high or low neutralizing antibody concentration were fit to various feature groups and for various cell types. For the CD4 and pooled TRB receptor repertoires, binding capacities as well as fuzzy matching features with tolerances 0, 1, …, 10 to the MIRA-identified CD4 clones from Nolan et al. [23] were used. Another set of models was derived from these by adding a mechanism at the end of feature selection that aggregates the selected features into their sums. For the pooled IGH, TRB, and TRD as well as the IGM, non- IGM (predominately IGG), CD4 TRB and CD8 TRB repertoires models are fit on the following feature sets:

- CDR3 length frequencies, summarized by 3 features: mean, variance and skewness;

- diversity, with Recon [55] (https://github.com/ArnaoutLab/Recon) being used to correct Hill $D_q$ numbers for $q = 0, 1, …, \infty$ to correct for missing species;

- J-gene frequencies (with only 6 J genes, no further feature selection was required);

- V-gene frequencies for select V genes as described above;

- Baseline/clinical features: age, sex, days since infection (runs of positive COVID-19 PCR tests successively within 28 days of each other and not interrupted by negative tests are considered infected periods; to account for incubation of the virus prior to taking the test, the start date of an infection is predicted as 4 days before the first positive test in the corresponding run of tests; when a negative test was performed within those 4 days, that test's date is considered the infection start date; for the model, the predicted start date of the most recent infection before sample collection was used, or 0 if the subject was not infected), and days since vaccination (the number of days between sample collection and most recent vaccination on record).

The machine learning framework was set up as follows. For each feature group, 700 replicate performances were measured via repetition of 7-fold cross-validation 100 times, each time choosing a different split of the data into 7 folds at random. For each replicate, 10-fold cross- validation was used to tune hyperparameters via Bayesian optimization. For each model fit, the training data was standardized, then underwent principal component analysis, and finally was used to train an L2-regularized regression. There were two tuned hyperparameters: regularization strength (with a log-uniform search space distribution between $10^2$ and $10^8$) and the amount of variance to be explained by chosen principal components (with uniform search space distribution between 0.50 and 0.99; e.g., if the value was 0.75 and the first four PCs account for 75% of variance, these four PCs would be chosen). For feature sets relating to similarity—binding capacities, fuzzy-matching features at various tolerances, and their aggregated versions—and for V-gene features, feature selection was performed on the training data before standardization for each model fit. To facilitate avoidance of train-test leakage, the mechanisms are implemented in the pymmunomics python package to fit into the popular scikit-lean API framework.

## Supporting information

**S1 File. Supplementary Figures S1-S15 and Supplementary Tables S1-S8.**
(DOCX)

## Author contributions

**Conceptualization:** Hamid Mirebrahim, Hosseinali Asgharian, Florian Rubelt, Ramy Arnaout.

**Data curation:** Jasper Braun, Elliot D. Hill, Hamid Mirebrahim, Hosseinali Asgharian, Florian Rubelt, Ramy Arnaout.

**Formal analysis:** Jasper Braun, Elliot D. Hill, Hamid Mirebrahim, Hosseinali Asgharian, Florian Rubelt, Ramy Arnaout.

**Funding acquisition:** Hamid Mirebrahim, Hosseinali Asgharian, Florian Rubelt, Ramy Arnaout.

**Investigation:** Jasper Braun, Elliot D. Hill, Elisa Contreras, Michie Yasuda, Alexandra Morgan, Sarah Ditelberg, Ethan Winter, Cody Callahan, Gabrielle Mazzoni, Andrea Kirmaier, Ghee Rye Lee, Hamid Mirebrahim, Hosseinali Asgharian, Dilduz Telman, Sanjucta Dutta, Florian Rubelt, Ramy Arnaout.

**Methodology:** Jasper Braun, Elliot D. Hill, Andrea Kirmaier, Hamid Mirebrahim, Hosseinali Asgharian, Florian Rubelt, Ramy Arnaout.

**Project administration:** Hamid Mirebrahim, Hosseinali Asgharian, Florian Rubelt, Ramy Arnaout.

**Resources:** Jasper Braun, Elliot D. Hill, Elisa Contreras, Michie Yasuda, Alexandra Morgan, Sarah Ditelberg, Ethan Winter, Cody Callahan, Gabrielle Mazzoni, Hamid Mirebrahim, Hosseinali Asgharian, Dilduz Telman, Ai-Ris Y. Collier, Dan H. Barouch, Stefan Riedel, Florian Rubelt, Ramy Arnaout.

**Software:** Jasper Braun, Elliot D. Hill, Alexandra Morgan, Hamid Mirebrahim, Hosseinali Asgharian, Stefan Riedel, Florian Rubelt, Ramy Arnaout.

**Supervision:** Hamid Mirebrahim, Hosseinali Asgharian, Florian Rubelt, Ramy Arnaout.

**Validation:** Jasper Braun, Elliot D. Hill, Elisa Contreras, Michie Yasuda, Alexandra Morgan, Hamid Mirebrahim, Hosseinali Asgharian, Dilduz Telman, Florian Rubelt, Ramy Arnaout.

**Visualization:** Jasper Braun, Elliot D. Hill, Ghee Rye Lee, Hamid Mirebrahim, Hosseinali Asgharian, Florian Rubelt, Ramy Arnaout.

**Writing – original draft:** Jasper Braun, Elliot D. Hill, Hamid Mirebrahim, Hosseinali Asgharian, Florian Rubelt, Ramy Arnaout.

**Writing – review & editing:** Jasper Braun, Elliot D. Hill, Andrea Kirmaier, Hamid Mirebrahim, Hosseinali Asgharian, Florian Rubelt, Ramy Arnaout.

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
