## [Decision Letter · Decision Letter 0]

16 Dec 2025

PONE-D-25-52234
Contrasting Effects of SARS-CoV-2 Vaccination vs. Infection on Antibody and TCR Repertoires
PLOS One

Dear Dr. Arnaout,

Thank you for submitting your manuscript to PLOS ONE. After careful consideration, we feel that it has merit but does not fully meet PLOS ONE’s publication criteria as it currently stands. Therefore, we invite you to submit a revised version of the manuscript that addresses the points raised during the review process.

We look forward to receiving your revised manuscript.

Kind regards,

Swati Jaiswal

Academic Editor

PLOS One

Journal Requirements:

https://journals.plos.org/plosone/s/file?id=ba62/PLOSOne_formatting_sample_title_authors_affiliations.pdexf

“HHS | NIH | National Institute of Allergy and Infectious Diseases (NIAID):Jasper Braun,Elliot D. Hill,Elisa Contreras,Michie Yasuda,Alexandra Morgan,Sarah Ditelberg,Ethan Winter,Cody Callahan,Gabrielle Mazzoni,Andrea Kirmaier,Ramy Arnaout R01AI148747; HHS | NIH | National Institute of Allergy and Infectious Diseases (NIAID):Jasper Braun,Elliot D. Hill,Elisa Contreras,Michie Yasuda,Alexandra Morgan,Andrea Kirmaier,Ramy Arnaout R01AI148747-SI; Massachusetts Life Sciences Center (MLSC):Jasper Braun,Elliot D. Hill,Elisa Contreras,Michie Yasuda,Alexandra Morgan,Sarah Ditelberg,Ethan Winter,Cody Callahan,Gabrielle Mazzoni,Andrea Kirmaier,Hamid Mirebrahim,Hosseinali Asgharian,Dilduz Telman,Stefan Riedel,Sanjucta Dutta,Florian Rubelt,Ramy Arnaout”

“This work was supported by the National Institutes of Health (NIH) National Institute for Allergy and Infectious Diseases (NIAID) R01AI148747; the Massachusetts Life Sciences Center Bits to Bytes program; the Extreme Science and Engineering Discovery Environment, which is supported by NSF Grant ACI-1548562; and the Comet supercomputer at the San Diego Supercomputer Center, which is supported by NSF Grant ACI-1341698.”

“HHS | NIH | National Institute of Allergy and Infectious Diseases (NIAID):Jasper Braun,Elliot D. Hill,Elisa Contreras,Michie Yasuda,Alexandra Morgan,Sarah Ditelberg,Ethan Winter,Cody Callahan,Gabrielle Mazzoni,Andrea Kirmaier,Ramy Arnaout R01AI148747; HHS | NIH | National Institute of Allergy and Infectious Diseases (NIAID):Jasper Braun,Elliot D. Hill,Elisa Contreras,Michie Yasuda,Alexandra Morgan,Andrea Kirmaier,Ramy Arnaout R01AI148747-SI; Massachusetts Life Sciences Center (MLSC):Jasper Braun,Elliot D. Hill,Elisa Contreras,Michie Yasuda,Alexandra Morgan,Sarah Ditelberg,Ethan Winter,Cody Callahan,Gabrielle Mazzoni,Andrea Kirmaier,Hamid Mirebrahim,Hosseinali Asgharian,Dilduz Telman,Stefan Riedel,Sanjucta Dutta,Florian Rubelt,Ramy Arnaout”

Reviewers' comments:

Reviewer's Responses to Questions

**Comments to the Author**

1. Is the manuscript technically sound, and do the data support the conclusions?

Reviewer #1: Partly

Reviewer #2: Partly

2. Has the statistical analysis been performed appropriately and rigorously?

Reviewer #1: Yes

Reviewer #2: I Don't Know

3. Have the authors made all data underlying the findings in their manuscript fully available?

Reviewer #1: Yes

Reviewer #2: Yes

4. Is the manuscript presented in an intelligible fashion and written in standard English?

Reviewer #1: Yes

Reviewer #2: Yes

5. Review Comments to the Author

Reviewer #1: Overall the data is valuable and the background is well-explained. However, the machine learning section for predicting NAb titer is relatively simplistic and lacks sufficient clarity. In particular, it is not clear how features based on exact and fuzzy sequence matching are generated from the repertoire data. It would be helpful to include a clear framework diagram illustrating the overall modeling pipeline, along with a small example of input data to show what the features actually look like. This would greatly improve understanding of how repertoire data is transformed into model inputs.

The use of NAbs as a functional readout makes sense for BCR, but it does not directly reflect whether TCR repertoires undergo antigen-specific changes. Some individuals may have strong T-cell responses but weak antibody responses, especially mild or asymptomatic cases or in some vaccinated individuals.

Figure 1 raises questions—NAb titers appear higher in vaccinees than infectees, particularly in younger individuals, and the difference is significant. Is this biologically reasonable? More explanation is needed.

Some references only include DOIs without journal names. Please ensure references are complete.

Reviewer #2: In this study, Braun et al attempt to identify CDR V and J codon usage and length from B and T cells isolated from patients infected with SARS-CoV-2 or immunized with the vaccine.

The paper is very complicated, and the presentation of the data is confusing with no clear biological insight.

Comments

In the abstract line 31-33: the authors state that: “Vaccination and infection have an effect on non-productively recombined IGHs, suggesting an effect that precedes clonal selection.”

This statement is confusing. It is better to say what the effect is rather than stating that there is an effect.

In the abstract line 33-36: the statement “We found that repertoires’ binding capacity to known SARS-CoV-2 specific CD4+ TCRB performs as well as the best hand tuned approximate or fuzzy matching at predicting a productive level of Nabs, while also being more robust to repertoire sample size and not requiring hand tuning.”

The statement is not clear. I recommend they rewrite it and explain what repertoires’ binding capacity they are referring to? T cell or B cells?

In the abstract line 38: What do they mean by surprising, subtle, and diffuse?

Major comments:

1. What is the biological or immunological relevance of the findings? This is not clear in the manuscript. It is unclear what the biological significance of all their bioinformatic analysis is. Can they predict what IGH and TCRB are responsible for the most neutralizing Abs?

2. They are insinuating that BCR/TCR rearrangement is affected by Ags, while the 2 processes are known to be independent. They do not provide genetic or biologic proof for that, other than there is an increase in non-productive receptor recombination by sequencing. Does infection decrease NHEJ enzymes and TdT, which are important for receptor modification at the junctions and their fusion?

3. The results has a lot of educational information that is too complex to understand by immunologists. It should be focused on the results and their significance.

4. The discussion is too long and repeats and expands concepts that are introduced in the introduction. It should be focused on discussing their results and compare/contrast them to other studies in the literature.

5. The authors obtained their DNA from B and T cells isolated from peripheral blood. B cells with longer IGH CDR3s in non-productive joins are negatively selected because they do not express a productive BCR, and, thus, do not egress out of the bone marrow into the periphery. How do the authors explain obtaining them from peripheral blood? Could they be sequencing the B cells that have one non-productive allele, and thus reinitiated VDJ recombination resulting in a productive recombination of the second allele? In that case, their data is a mix of sequences from productive and non-productive alleles sometimes originating from the same B cell clones.

6. They only focus on type of CDRs and length of mismatch at the joins, but they do not account for the type of nucleotide at the joins that results in a different amino acid and thus a different Ab variable domain.

7. Line 213: do they mean “Thus selection for “longer” CDR3s in…”

8. Line 230: correct the English.

9. Line 282-283: The statement: “Thus, fuzzy matching was shown to not be a robust measure of repertoire content in this study.” is written in poor English. In addition, the authors write this complex paragraph to explain the principle of fuzzy matching to readers who are not expert in the field. Yet at the end they find negative correlation with fuzzy matching as a measure of robust repertoire analysis. Why not omit this information and just mention using the binding capacity method?

6. PLOS authors have the option to publish the peer review history of their article (what does this mean?). If published, this will include your full peer review and any attached files.

Reviewer #1: No

Reviewer #2: No

---

## [Author Response · Author response to Decision Letter 1]

1 Jan 2026

Please see attachment marked Response to Reviewers, as requested.

---

## [Editor Report · Decision Letter 1]

12 Feb 2026

Contrasting Effects of SARS-CoV-2 Vaccination vs. Infection on Antibody and TCR Repertoires

PONE-D-25-52234R1

Dear Dr. Arnaout,

We’re pleased to inform you that your manuscript has been judged scientifically suitable for publication and will be formally accepted for publication once it meets all outstanding technical requirements.

Kind regards,

Swati Jaiswal

Academic Editor

PLOS One

---

## [Editor Report · Acceptance letter]

PONE-D-25-52234R1

PLOS One

Dear Dr. Arnaout,

I'm pleased to inform you that your manuscript has been deemed suitable for publication in PLOS One. Congratulations! Your manuscript is now being handed over to our production team.

Kind regards,

on behalf of

Dr. Swati Jaiswal

Academic Editor

PLOS One